# Integrated environmental and clinical surveillance for the prevention of acute respiratory infections (ARIs) in indoor environments and vulnerable communities (Stell-ARI): Protocol

**Annalaura Carducci**[1], **Guglielmo Arzilli**[2], **Nebiyu Tariku Atomsa**[1]*, **Giulia Lauretani**[1], **Marco Verani**[1], **Francesco Pistelli**[3], **Lara Tavoschi**[2], **Ileana Federigi**[1], **Marco Fornili**[4], **Davide Petri**[4], **Tommaso Lomonaco**[5], **Claudia Meschi**[3], **Alessandra Pagani**[1], **Antonello Agostini**[2], **Laura Carrozzi**[3], **Laura Baglietto**[4], **Daniela Paolotti**[6], **Ciro Cattuto**[6], **Lorenzo Dall'Amico**[6], **Caterina Rizzo**[2]

**1** Hygiene and Environmental Virology, Department of Biology, University of Pisa, Pisa, Italy, **2** Department of Translational Research and New Technologies in Medicine and Surgery, University of Pisa, Pisa, Italy, **3** Department of Surgical, Medical and Molecular Pathology and Critical Care Medicine, University of Pisa, Pisa, Italy, **4** Department of Clinical and Experimental Medicine, University of Pisa, Pisa, Italy, **5** Department of Chemistry and Industrial Chemistry, University of Pisa, Pisa, Italy, **6** Italian Institute for Scientific Interchange, ISI Foundation, Turin, Italy

* nebiyu.atomsa@phd.unipi.it

## Abstract

The epidemiological relevance of viral acute respiratory infections (ARIs) has been dramatically highlighted by COVID-19. However, other viruses cannot be neglected, such as influenza virus, respiratory syncytial virus, human adenovirus. These viruses thrive in closed spaces, influenced by human and environmental factors. High-risk closed communities are the most vulnerable settings, where the real extent of viral ARIs is often difficult to evaluate, due to the natural disease progression and case identification complexities. During the COVID-19 pandemic, wastewater-based epidemiology has demonstrated its great potential for monitoring the circulation and evolution of the virus in the environment. The "Prevention of ARIs in indoor environments and vulnerable communities" study (Stell-ARI) addresses the urgent need for integrated surveillance and early detection of ARIs within enclosed and vulnerable communities such as long-term care facilities, prisons and primary schools. The rapid transmission of ARIs in such environments underscores the importance of comprehensive surveillance strategies to minimise the risk of outbreaks and safeguard community health, enabling proactive prevention and control strategies to protect the health of vulnerable populations. This study consists of designing and validating tools for integrated clinical and environmental-based surveillance for each setting, coupled with analytical methods for environmental matrices. The clinical surveillance involves specialized questionnaires and nasopharyngeal swabs for virus identification, while the environmental surveillance includes air and surface microbiological and chemical monitoring, and virological analysis of wastewater. Integrating this information and the collection of behavioural and environmental risk

**Data Availability Statement:** No datasets were generated or analysed during the current study. All relevant data from this study will be made available upon study completion.

**Funding:** This project is funded by the National Government of Italy through the National Recovery and Resilience Plan (NRRP) 'Italy Tomorrow', which is part of the Next Generation EU (NGEU) programme. The NRRP aims to address the economic and social impacts of the SARS-CoV-2 pandemic crisis while addressing structural weaknesses in the Italian economy and promoting ecological and environmental transition. The Tuscany Health Ecosystem (THE), coordinated by the University of Florence, aligns with the NRRP's focus on 'Life Sciences', fostering growth in Tuscany's life sciences sector through collaboration between public and private entities to address innovation needs. THE comprises 10 Spokes, with our study falling under Spoke 2 (Preventive and Predictive Medicine), led by the University of Pisa's Department of Biology, in collaboration with other departments. In accordance with the NRRP, the Research and Innovation Programme "THE - Tuscany Health Ecosystem" has a 36-month eligibility period, commencing on 01/12/2022 and extending no later than 28/02/2026. Investment Line 1.5 allocates funding for the creation and/or strengthening of 12 Innovation Ecosystems (THE) across the national territory. This specific project falls under Spoke 2 (Preventive and Predictive Medicine), with project code I53C22000780001 with a budget of 1,308,657.86 €. The funders (The Italian Government) had no involvement in the study design, data collection and analysis, decision to publish, or manuscript preparation.

**Competing interests:** The authors have declared that no competing interests exist.

**Abbreviations:** ACAQ, Acute Clinical Assessment Questionnaire; ARIs, Acute Respiratory Infections; BCQ, Basal Clinical Questionnaire; $CO_2$, Carbon Dioxide; COVID-19, Coronavirus Disease 2019; CUP, Unique Project Code; EQ, Environmental Questionnaire; GCP, Good Clinical Practice; HAdV, Human Adenovirus; Inmates, People Who Live In prison; IVAB, Influenza A & B virus; LTCFs, Long-Term Care Facilities; NRRP, National Recovery and Resilience Plan; PCR, Polymerase Chain Reaction; $PM_{2.5}$, Particulate matter (atmospheric particles—2.5 µm); PPE, Personal Protective Equipment; QMRA, Quantitative Microbial Risk Assessment; RFID, Radio Frequency Identification; RSV, Respiratory Syncytial Virus; RT-PCR, Real time PCR; SARS-CoV-2, Severe Acute Respiratory Syndrome CoronaVirus 2; Staff, all workers in

factors into predictive and risk assessment models will provide a useful tool for early warning, risk assessment and informed decision-making. The study aims to integrate clinical, behavioural, and environmental data to establish and validate a predictive model and risk assessment tool for the early warning and risk management of viral ARIs in closed and vulnerable communities prior to the onset of an outbreak.

---

## Introduction

Viral Acute Respiratory Infections (ARIs) are a broad spectrum of illnesses caused by different viruses that can infect both the upper and lower respiratory tracts. These diseases are highly contagious and can spread through various mechanisms, such as direct contact, fomites, aerosol and droplets of saliva or mucus expelled by infected individuals during sneezing, coughing, speaking, and singing [1].

The emergence of respiratory viral diseases continues to threaten global public health security [2]. Even with the efforts of public health strategies to limit infections through widespread testing and social distancing, COVID-19 has led to millions of infections and fatalities across the globe, leading to substantial social and economic disruptions [3, 4]. Moreover, the global incidence of ARIs caused by influenza virus and respiratory syncytial virus (RSV) should not be overlooked. It is estimated that influenza virus causes 39.1 million episodes of acute lower respiratory tract infection (95% uncertainty interval 30.5–48.4) and 58,200 deaths (44,000–74,200) per year. Similarly, RSV causes 24.8 million cases (19.7–31.4) and 76,600 deaths (55,100–103,500) per year [5]. Other viruses, such as other coronavirus, metapneumovirus, rhinovirus, adenovirus can also contribute to a significant disease burden [6].

Usually, the dynamics of ARIs in a population are described through clinical surveillance, which requires specific procedures to monitor and collect data on the incidence, prevalence, case severity, and fatalities. Clinical surveillance strategies focused on infection tracking, which are symptom-based, should be revised and integrated, as they tend to underestimate cases due to the presence of asymptomatic individuals who may not be identified nor promptly notified [7]. This approach often captures only a fraction of viral disease cases (hospitalised patients or laboratory-diagnosed cases), representing only the tip of the iceberg. Therefore, limitations such as accessibility and expenses, as well as difficulties in identifying asymptomatic individuals, can be challenging [8].

Wastewater Based Epidemiology (WBE), initially introduced for the poliovirus surveillance, has shown its high potential, especially during the COVID-19 pandemic, making it a very promising tool for monitoring various infections as well as for antimicrobial resistance [9, 10]. In the COVID-19 pandemic scenario, it has been a valuable instrument for understanding the full extent of viral circulation and as an alert system during a rise of SARS-CoV-2 infection [7, 11]. Most people infected with SARS-CoV-2 are asymptomatic, but they shed viral particles, particularly in faeces and other body fluids, which ultimately reach sewage systems. Therefore, detecting the genome of SARS-CoV-2 in urban wastewater has been useful in capturing the full extent of the disease at a community level, although the viral genome in this matrix is not associated with infectivity [12].

WBE can detect pathogen circulation before the onset of cases, making it a valuable early warning tool and facilitating the timely identification of new variants. This approach has been extensively used, not only in large urban areas but also in smaller communities, to identify the

LTCF, Prison or School; SVOCs, Semi-volatile organic compounds; TB, Tuberculosis; TD-GC-MS, Thermal desorption coupled with gas chromatography and mass spectrometry; THE, Tuscany Health Ecosystem; VOCs, Volatile organic compounds (volatile organic compounds); WBE, Wastewater-Based Epidemiology.

initial introduction of pathogens, even in the absence of symptomatic infections, enabling the implementation of prevention and control measures [13].

Both types of surveillance have strengths and limitations and may be susceptible to bias. A surveillance system integrating data from multiple sources may overcome these challenges and better describe the epidemiological scenario of ARIs. This integration is particularly useful in enclosed settings, especially for individuals who may experience vulnerabilities, increasing their risk of developing ARIs and/or experience more serious clinical consequences, such as age (children or the elderly), underlying health conditions, or socio-economic disadvantage.

While WBE studies have been carried out in LTCFs [14] and schools [15] to implement prevention measures, an integrated surveillance approach is lacking. Data are even more limited for prisons [16], which pose a high risk for ARIs due to the prevalence of additional risk factors (e.g. substance use, including tobacco, suboptimal individual hygiene) and environmental factors such as overcrowding and infrastructural deficiencies (e.g. ventilation systems).

In these contexts, there is also a strong need for risk assessments to effectively identify the epidemiological conditions to timely undertake preventive and control measures. The environmental spreading of pathogens causing respiratory infections, which was analysed in detail during the COVID-19 pandemic, can be represented as a chain of events, each affecting the ultimate risk for further transmission. These events include droplet/aerosol emission, their environmental dynamics (falling or remaining suspended), the viability of pathogens in different matrices and conditions, the ways and duration of exposure, and the infection/disease according to dose-response relations [1, 17].

Most respiratory viruses like SARS-CoV-2 spread through aerosols and fomites, remaining viable in the air and on surfaces [18, 19]. Environmental characteristics, particularly air quality and living/working conditions, significantly influence disease transmission rates, which may be higher in enclosed or semi-enclosed crowded settings with limited air circulation [20]. Data on contamination of vehicles (namely air and surfaces) and on pathogen survival contribute to understanding the spreading pathways and assessing infection risks through Quantitative Microbial Risk Assessment (QMRA) [21]. Beyond environmental contamination, frequent close contacts, typical of school-age children, are a further element in the spread of infectious diseases. Furthermore, individual risk factors and underlying chronic illnesses are often associated with frail conditions, facilitating infections and their severity.

Integrating clinical surveillance with wastewater and environmental analysis, and taking into account individual and behavioural risk factors, can aid designing more effective prevention strategies. In order to achieve this goal, the infectious diseases we target should meet several criteria. These include having a clear case definition supported by diagnostic microbiological confirmation, and identifiable environmental pathways that can be traced using reliable microbiological detection methods. Among ARIs, those caused by SARS-CoV-2, influenza viruses, and RSV are of major concern for public health, and they can be effectively monitored through clinical and environmental surveillance methods. Therefore, these viruses have been selected for our study, along with Human Adenovirus, which are widespread and often used for QMRA in water, air and on surfaces [22].

## Method/Design

### Aim

The general aim of this protocol is to create an integrated epidemiological, clinical and environmental surveillance system for ARIs and their determinants within community settings in order to plan prevention strategies and intervention measures to minimise infection spread.

This system will be designed and piloted in selected close settings, where people spend long period of time, often in crowded and poorly ventilated spaces.

Combining risk assessment and early warning could enhance the opportunities and modalities to implement effective prevention and control measures. Yet, the development and validation of such a system requires the integration of clinical-epidemiological and environmental data, which can only be achieved by conducting multiple streams of data collection in a parallel and coordinated manner.

Therefore, specific objectives of the study are:

Obj. 1. Set up sentinel clinical surveillance systems in selected closed settings to a) detect individual and social behaviours related to ARIs, b) identify individual risk factors, including comorbidities and close contacts, c) recognize early signs and symptoms, following diagnostic criteria for specific respiratory diseases and their progression d) detect etiological agents through confirmatory sampling and analysis.

Obj. 2. Establish sentinel environmental surveillance in selected closed environment to a) collect data on environmental parameters (e.g., space, crowding, microclimate conditions and ventilation), b) monitor indoor air quality, including air pollutants, microbiological indicators, and viral pathogens and c) conduct virological monitoring of wastewater.

Obj. 3. Integrate clinical, behavioural, and environmental data to develop early warning system, risk assessment (QMRA) and risk prediction models that can be replicable and transferable to similar settings at the regional, national, and international levels.

## Study design

This is an open, multi-setting, epidemiological cohort, and environmental monitoring study including participants recruited from one LTCF (residents and staff), one remand prison (people living in prison and staff), one primary school (students and teachers). The project spans three years, from December 2022 to December 2025, and is articulated in three distinct phases:

- Phase 1 (Dec 2022—Nov 2023) developing study protocol, enrolling facilities, developing data collection instruments (questionnaires and analytical methods for clinical and environmental analyses), obtaining ethics committee approval and engaging target communities (or study participants);

- Phase 2: (Dec 2023—Nov 2024) conducting field study data collection;

- Phase 3: (Dec 2024—Nov 2025) processing collected data, developing and validating models, creating information materials and disseminating findings.

The study workflow is shown in **Fig 1**.

For each recruited population and site, the following surveys will be carried out:

Clinical surveillance: semi-structured pseudonymised questionnaires will be administered to participating individuals and facility staff to detect baseline risk factors and individual symptoms during acute episodes. In case of suspected ARIs, nasopharyngeal swabs will be conducted to identify responsible viruses (for the chosen viral targets) by means of biomolecular Polymerase Chain Reaction (PCR) tests. Data on close contacts will only be collected for students during school hours, using proximity sensors (tags) based on Radio Frequency Identification (RFID) technology allowing pseudonymised detection of close contacts between individuals within the school environment [23].

Environmental surveillance: semi-structured questionnaires will be administered to facility managers to survey the characteristics of the premises and the frequency of attendance. Periodic monitoring of microclimate parameters and the chemical-microbiological quality of the

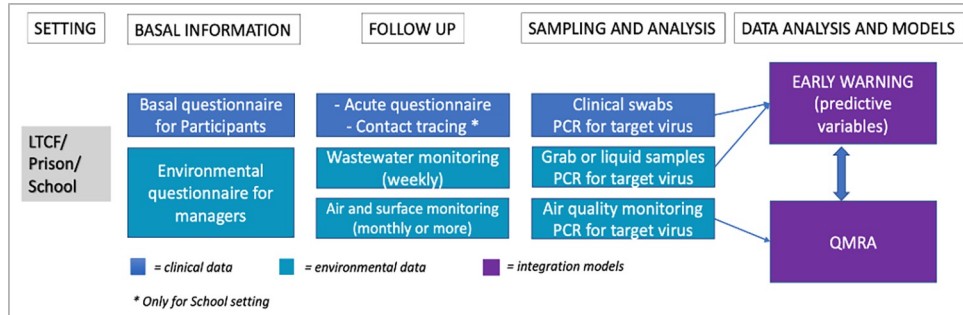

**Fig 1. Scheme of the study workflow: Clinical data in blue squares, environmental data in light blue squares, integration models in purple.**

air will be conducted. Target viruses in the air and on surfaces of common areas will be detected using biomolecular PCR tests. Wastewater will be virologically monitored through sample concentration and biomolecular analysis.

Finally, data processing and model development will integrate clinical, behavioural, and environmental data to create to develop and validate models for ARIs alerts and risk assessment in closed communities.

## Study setting and participants

The study encompasses three distinct settings:

- a LTCF located in the province of Pisa, Italy which provides care for elderly individuals, offering medium-complexity and permanent/temporary care with 71 beds divided into four units. Residents over the age of 65 and all carers will be included in the study. The study will include residents over the age of 65 and all carers. Residents have multiple health conditions, including chronic diseases, degenerative diseases, physical disabilities, and varying levels of functional autonomy. This includes those who require assistance with daily activities such as mobility, personal hygiene, mealtimes, and other routine activities. Residents come from diverse socio-economic statuses and educational backgrounds. Inclusion criteria encompass all genders and origins, residents, and staff members over 18 years who have provided informed consent. Exclusion criteria include individuals who do not provide informed consent or are unable to comprehend instructions or respond to questions.

- A remand prison in the province of Lucca, Italy, housing more than 75 inmates. The study will include all detained individuals aged over 18 and all the staff of the. People in detention in this facility have short sentences (less than 5 years) or are awaiting judgement, with various clinical conditions such as chronic diseases, infectious diseases and other known or diagnosed medical conditions. The individuals included will have different educational and socio-economic levels, with a high proportion of foreign nationals reflecting the population characteristics. Inclusion criteria includes all gender identities, country of origin/nationality, and those who have signed informed consent. Exclusion criteria include individuals who do not provide informed consent, or those unable to comprehend instructions or respond to questions.

- a primary school in Pisa, Italy, attended by students between the ages of 5 and 11 years and primary school staff members. The study will include all individuals regardless of health status, language, or nationality. Only students with written informed consent will be enrolled. All adult individuals who do not provide informed consent, students for whom parental

informed consent was not obtained, or individuals incapable of understanding instructions or responding to questions will not be included in the study.

## Data collection

**Data collection tool and variables.**    During the second phase of the study, data will be collected through questionnaires and analytical procedures. This integrated approach will yield a comprehensive dataset, enhancing the study's clinical and environmental assessment as a whole.

*1. Questionnaires.* The questionnaires have undergone a meticulous design process, incorporating both newly formulated sections and standardized or adapted questions from previously employed surveys in published studies [24–28]. To ensure their effectiveness, these questionnaires have been subjected to multiple revisions by experts across various departments within the field. This comprehensive review aimed to guarantee that the questions align with the constructs intended for assessment. Upon the initial phase of data collection, the robustness of these questionnaires will be further evaluated. Internal consistency will be assessed using the Cronbach's alpha test, ensuring that the questions within each questionnaire measure the intended constructs reliably. This rigorous evaluation process is integral to maintaining the quality and precision of the data we aim to collect.

The following information will be collected through the questionnaires:

- Environmental characteristics of the settings (including surfaces, volumes, ventilation, and common areas), daily activities performed, and hygiene practices within the facilities

- Individual participant characteristics including general attributes, existing medical conditions such as respiratory infections and allergies, vaccination history, exposure to risk factors, socio-economic factors, any acute illnesses experienced during the study period, contact history with other individuals, and their consent for potential nasopharyngeal swab testing.

*1.1 Environmental questionnaire*: The purpose of this questionnaire is to understand the current hygiene policies and practices within the facilities and to identify factors that might influence the risk of ARIs.

The environmental questionnaire (EQ) is specifically designed for the managers of the various settings. It evaluates the environmental structure and conditions, as well as individual and group behaviour and practices within these settings.

The questionnaire will cover several aspects, including personal and environmental hygiene practices, cleaning and disinfection of common areas, ventilation, air quality and daily activities. It will consist of a series of multiple-choice and open-ended questions to cover different environmental factors in order to obtain specific information on participants behaviour, their movements and contacts within the facility. In addition, the questionnaire will assess the efficacy of existing policies on preventing the spread of respiratory viruses, as well as the facility's ability to adopt any policy changes in a timely and effective manner. The EQ will be administered once at the beginning of the survey to the facility managers and will be administered again later if there are substantial changes in the facility. The questionnaire will take approximately 8 minutes to complete.

*1.2 Baseline clinical questionnaire*: The objective of the baseline clinical assessment questionnaire (BCQ) is to provide a complete and accurate picture of the participant's clinical status, identifying potential risk factors and individuals susceptible to developing complications from ARIs. The questions have been crafted by combining established question formats,

including inquiries about pre-existing respiratory symptoms, allergic history, and risk factors [24–26], with the integration of newly devised questions designed to align with the objectives of our project.

The BCQ is intended for all participants and consists of a series of multiple-choice and open-ended questions covering different aspects of the participant's medical history, such as the presence of previous illnesses, risk factors for respiratory infections, previous vaccinations, respiratory and allergic symptoms, socio-economic factors, and exposure to risk factors such as smoking and alcohol consumption. The data collected by the BCQ will allow a baseline assessment of the subject's condition and evaluate a possible correlation with respiratory virus infections. The BCQ will only be administered once at the beginning of the survey, or when a new participant joins the study (e.g. transfer to the facility). The questionnaire will take approximately 12 minutes to complete.

*1.3 Acute clinical assessment questionnaire*: The objective of the Acute Clinical Assessment Questionnaire (ACAQ) is to collect information that will allow us to monitor the evolution of the subject's condition over time, to identify possible clusters of respiratory infections within the community environment, and to take the necessary prevention and control measures to prevent the spread of infection. The questions have been formulated by merging established question formats pertaining to acute symptoms and manifestations [27, 28] with newly integrated inquiries that specifically align with the objectives of our project.

The ACAQ is an essential tool for the early identification of any symptoms associated with ARIs and for monitoring the evolution of the subject's condition over time. It is designed to identify all participants presenting with respiratory symptoms, such as cough, sore throat, difficulty breathing or fever, and is intended to provide a detailed and timely clinical assessment of participants. The questionnaire consists of a series of specific questions regarding the subject's current symptoms, the duration of symptoms, any current therapies and contact with other individuals who have experienced similar symptoms. The completion of each questionnaire will take approximately 6 minutes.

*1.4 Methods of questionnaire administration*: The questionnaires will be administered with different methods and supports based on the target. The method of administration, support and frequency are summarised in **Table 1** for each setting and target population.

For individuals who are unable to fill out the questionnaire due to health or social reasons (such as residents in LTCFs and inmates), the questionnaires will be administered during a face-to-face interview. Specifically trained interviewers will ask the questions and record the answers on paper forms or tablets, depending on the setting. For other participants (such as parents of students and staff), the questionnaires will be self-administered in paper form, on tablet, or through the web portal Influweb (for primary school). These questionnaires are designed to be simple, easy to understand and fill, and are accompanied by short and clear instructions. They will be distributed along with a brief explanation of the survey's objectives, to motivate the participation and cooperation of participants. Completed paper questionnaires will be collected in a dedicated closed box to guarantee the anonymity of the participant and will be transported to the Department of Biology Department of the University of Pisa. For tablet or web portal administration (https://influweb.org/welcome), study participants will receive a letter of introduction and detailed instructions.

For the school setting, questionnaire data is collected through the platform of Influweb, which is an Italian specialised digital platform designed for the efficient administration of surveys and questionnaires related to data on Influenza-Like Illness (ILI). It is a project supported by Fondazione ISI, Fondazione CRT and from the European projects Epipose and PANDEM-2, part of the EU Horizon 2020 program. It provides a user-friendly interface for data collection and analysis, making it particularly valuable in research and public health initiatives. This

**Table 1. Types and methods used for the questionnaires in the various settings of the Stell-ARI study.**

| Type of questionnaires | Setting | Target | Method of administration | Support | Frequency |
|---|---|---|---|---|---|
| Environmental Assessment Questionnaires | LTCF | Head of facility manager | Self-administered | Paper form | 1 time and in case of changes |
| | Prison | | | | |
| | School | | | | |
| Baseline clinical evaluation questionnaires | LTCFs | Residents | Interviewer-administered | Tablet | 1 time at the beginning and for new entries |
| | | Staff | Self-administered | | |
| | Prison | Inmates | Interviewer-administered | Paper form | |
| | | Staff | Self-administered | | |
| | School | Parents of students and Staff | Self-administered | 'Influweb' platform | |
| Acute Clinical Assessment Questionnaires | LTCF | Residents | Interviewer-administered | Tablet | In case of new occurrence of Symptoms |
| | | Staff | Self-administered | | |
| | Prison | Inmates | Interview | Paper form | |
| | | Staff | Self-administered | | |
| | School | Parents of students and Staff | Self-administered | 'Influweb' platform | |

technology enhances our ability to effectively detect and respond to ILI outbreaks, ultimately contributing to the prevention and control of these illnesses [29].

*2. Analytical procedures*. Analytical procedures will involve:

- Nasopharyngeal swabs in symptomatic subjects: detection of the etiological agents for confirmation of the infection

- Air analysis: assessment of microclimatic conditions, chemical indicators, microbiological indicators, and the presence of key pathogens in the air

- Surface Analysis: assessment of microbiological indicators and the presence of key pathogens on various surfaces

- Wastewater analysis: quantification of genomic copies of target viruses in wastewater samples.

- Proximity sensors (in school setting): detection of close encounters among students to analyse the risk of transmission within the school environment.

*2.1 Clinical sampling*: Biological material, such as surface cells, mucus, and secretions from the respiratory tract, will be collected using nasopharyngeal swabs. These diagnostic tests are designed to detect pathogenic microorganisms that cause upper respiratory tract infections. The swabs are minimally invasive, easy to perform, and painless. The sampling procedure involves using a thin, sterile, single-use wadded stick. This procedure will be carried out by the facility staff (nursing staff in the LTCFs and prison settings) and by parents and staff themselves in school setting, with the consent of the study subject or their guardian. To ensure proper execution and prevent sample contamination, appropriate personal protective equipment (PPE) will be used, as recommended by the Centers for Disease Control and Prevention (CDC) [30]. After collection, the swab will be placed in a sterile tube containing a transport medium. Following hand hygiene protocols, the collector will label the tube with a pseudonymised code and arrange for its transportation to the laboratory via a dedicated service.

*2.2 Environmental microbiological sampling*: To identify highly contaminated areas, an extensive distribution of "exposed plates" containing a non-selective medium will be deployed.

After a 6-hour exposure period, these plates will be incubated, and colony-forming units will be counted [31].

For air sampling, automatic samplers will be positioned in common areas to detect bacterial and viral contamination on a monthly basis. Active samplers will be employed, which draw in air and either impact it on a plate (e.g., SAS—Surface Air System sampler) or place it into a liquid (e.g., impinger sampler). In particular:

- Bacterial counts will be obtained by using an impactor sampler, with Rodac plates containing Plate Count Agar (PCA) or selective medium. Subsequently, the plates will be incubated.

- Viral analysis will be performed by using Impinger Sampler, consisting of an air inlet structure and a collection vessel filled with a liquid medium, Viral Transport Medium (VTM) [32].

Surface sampling will also be conducted in these areas on a monthly basis. Sterilized swabs will be swiped across 36 cm$^2$ areas [33]. For viral analysis, virus will be eluted from the swabs with 1 mL of 3% beef extract solution at pH 9, whereas for bacteriological analysis elution will be performed with 0.9% NaCl solution [34].

In wastewater analysis, the primary focus will be on target viruses. Trained personnel will perform weekly sampling at the exit point of each facility via a sewer drain outside the building. Necessary safety measures, including the use of PPE, will be strictly applied. A passive sampling method will be employed, as suggested for small communities with irregular sewage flow [35]: this involves immersing passive materials (electronegative filter membranes or gauze swabs) into the wastewater flow and leaving them for an extended duration [36]. Viruses will subsequently be eluted from these passive materials with 1 aliquot of sterile phosphate buffer solution. The solution will be centrifuged at 4500 rcf for 10 minutes to remove debris and then analysed using a commercial kit.

*2.3 Microbiological analyses*: Air and surface samples will be analysed for indicators of general/human microbial contamination (total bacterial load at both 22°C and 37°C, as well as Staphylococcus aureus and Escherichia coli) using conventional cultural methods. Specific respiratory viruses: Human Adenovirus (HAdV), Influenza Virus A and B (IVA and IVB), SARS-CoV-2, Respiratory Syncytial Virus Type A and Type B (RSVA and RSVB) will be detected using PCR techniques. The same RT-PCR protocols, summarised in **Table 2**, will be applied to both clinical and environmental samples. For wastewater samples, the amount of genome copies obtained will be adjusted on the basis of the sewage flow to estimate the daily viral emissions from the community.

**Table 2. PCR protocols used for the detection of viral targets.**

| Virus | Target region | Primers and probes | Sequences (5′-3′) | Reference |
|---|---|---|---|---|
| SARS-CoV-2 | ORF1ab region: nsp14 | 2297 CoV-2-F | ACA TGG CTT TGA GTT GAC ATC T | Verani et al., 2022 [37] |
| | | 2298 CoV-2-R | AGC AGT GGA AAA GCAT GTG G | |
| | | 2299 CoV-2-P | FAM—CAT AGA CAA CAG GTG CGC TC-MGBEQ | |
| HAdV | Hexon gene | AdF | CWT ACA TGC ACA TCK CSG G | Hernroth et al., 2002 [38] |
| | | AdR | CRC GGG CRA AYT GCA CCA G | |
| | | AdP1 | FAM—CCG GGC TCA GGT ACT CCG AGG CGT CCT – TAMRA | |
| IVA* | TaqManMicrobial Assays | | assay ID Vi99990011_po | Ahmed et al., 2023 [39] |
| IVB* | TaqMan Microbial Assays | | assay ID Vi99990012_po; | |
| RSVA* | TaqMan Microbial Assays | | assay ID Vi99990014_po | Ahmed et al., 2023 [39] |
| RSVB* | TaqMan Microbial Assays | | assay ID Vi99990015_po | |

*A commercial kit will be used for the detection of IVA, IVB, and RSV.

*2.4 Environmental chemical samples*: To identify the most contaminated sampling points within the three structures being studied, a preliminary chemical air characterization will be performed using diffusive samplers coupled with gas chromatography and mass spectrometry (TD-GC-MS). This protocol is able to monitor a wide range (C3-C30) of volatile organic compounds (VOCs) and semi-volatile organic compounds (SVOCs) [40]. The passive sampler will be placed at various sampling points for 6 hours allowing VOCs and SVOCs to be retained by the sorbent material of the sampler. After sampling, the diffusive bodies of the samplers will be hermetically sealed with Swagelock caps avoiding any external contamination. Routine chemical analysis will be carried out monthly by transferring an aliquot (1 L) of air samples with a suction pump into stainless steel absorbent systems packed with 250 mg of Tenax Gr 60/80 mesh. Prior to sampling, each absorbent tube will be spiked with 30 ng of labelled 8D-toluene to ensure an accurate quantification of the analytes. The result of the chemical analysis will contain information on the qualitative chemical composition of the air samples as well as the concentration levels of each contaminant found. An Arduino-based platform will be installed at the different sites to continuously monitor temperature, humidity, $CO_2$ levels, PM2.5, and total VOCs.

*2.5 Proximity sensors (students only)*: Proximity sensors gather the time-resolved proximity relationships between each sensor and the others within the system, along with information on the sensor's state of motion obtained through an on-board accelerometer in the device. This technology offers a promising approach for monitoring and managing social interactions, providing both the frequency and duration of contact. The sensors collect data, which will be stored internally in the devices (tags) and extracted at the end of data collection, via a physical connection between the sensors and the operators' computers [23]. All students who have given their consent will be required to wear the proximity devices every day in class for a period of 10 to 15 days.

The data collected will include patterns of close person-to person encounters, interaction dynamics between different groups, and pathways of environmental exposure, using diverse indicators. The high spatial and temporal resolution infrastructure will enable monitoring the number of individual contacts, encounter durations, cumulative interaction times, and encounter frequencies between any two individuals. For each pair, we will collect the following data: occurrence and frequency of each contact, time spent in each encounter, and cumulative contact duration per subject. In relation to the collected data, the following metrics will be obtained: number of distinct contacts, total number of contacts and cumulative contact time. In addition to representing a minimally invasive method of data collection, the tags require only minimal effort from participants. Proximity sensors will enable accurate, automated and pseudonymised detection of close encounters among students within the school environment.

## Follow-up and environmental monitoring

After the baseline survey, follow-up will be conducted to clinically assess the study subjects by means of ACAQ. As timing of administration cannot be predetermined, the occurrence of acute ARIs cases will be assessed on weekly bases in the LTCF and prison, through planned phone contacts with facility study reference personnel. For school setting, cases will be notified by parents through the Influweb platform. For each case a swab will be collected and the ACAQ will be administered.

Since the study population might change due to new entrants or exits during the follow-up, data collection will take into consideration these variations by calculating person/months of observation.

Both air and surface sampling will be carried out monthly, with the possibility of increasing the frequency in case of disease occurrence.

Wastewater samples will be collected weekly.

## Sample size

As this project aims to study several respiratory diseases, it is not feasible to calculate the sample size based on a specific prevalence.

The sample size needed to estimate a prevalence with a fixed margin of error depends on the unknown proportion, with a maximum size at a prevalence of 0.50: assuming this value, we obtain a conservative sample size of 380 to estimate a prevalence with a margin of error of 0.05 with a confidence of 95%.

To achieve this number of subjects, all eligible LTCF residents, people living in prison, students and facility staff who meet the inclusion criteria will be invited to participate in the study.

The number of environmental samples for microbiological and chemical analysis for each target setting and for three sampling points will be one per month for air and surface matrices (approximately 110 samples of air and 110 samples of surface). If clinical cases suspected of ARIs are detected, the number of environmental samples will be increased. Wastewater sampling will be performed monthly for each enrolled setting and for one sampling point (approximately 150 samples).

## Data management and storage

All collected data will be pseudonymised, and no sensitive information will be requested from recruited participants.

Pseudonymization will be performed by the LTCF, and prison staff based on investigator-provided unique identifiers lists. For the primary school, this process will be automatically performed by the Influweb platform, which will provide unique identifiers to parents/guardians.

The collected data will be stored in a secure computerised repository for analysis, with paper forms digitally converted. Data collected via the 'Influweb' platform and 'tag' devices will be managed directly by platform operators, who will extract the data by means of pseudonymised codes.

Electronic databases will be established and maintained to ensure data security, integrity and reliability, protecting against loss, alteration, or corruption.

The data will be processed in accordance with the provisions of the European Regulation for the processing of personal data no. 2016/679 and Legislative Decree of 30 June 2003 no. 196 ss.ii. as amended by Legislative Decree of 10 August 2018, no. 101 "Provisions for the adaptation of national legislation to the provisions of the GDPR", as well as the Deliberation of the Privacy Guarantor no. 52 of 24 July 2008 and subsequent updates.

The data, processed electronically, will be disseminated only in aggregate and anonymized form, such as through scientific publications, reports, and academic conferences.

## Statistical analysis and quantitative microbial risk assessment (QMRA)

**Statistical analysis.** Subject characteristics will be described by means of medians and interquartile ranges for numerical variables and absolute frequencies and percentages for categorical variables. The time course of the environmental variables will be presented through graphical displays.

Associations between infections detected via ACAQ and confirmed by clinical swabs (when necessary) and exposure factors detected by the QCB and the environmental air and surface

monitoring will be assessed employing logistic regression models taking potential confounders into account. Statistical methods and analysis algorithms will be used to examine the collected data by proximity sensors and identify contact patterns, interaction frequencies and other relevant metrics.

WBE data will be normalised and adjusted to reduce variations arising from excretion and sewage flow. Environmental data will be treated as continuous variables, with regression models featuring splines used to estimate the time course of the measured parameters. The relationship between ARIs incidence and environmental data will be explored via time series regression models. In the event of missing data, imputation models will be employed, and sensitivity analysis will evaluate the impact of outliers on results. All analyses will encompass the entire dataset or stratified by setting. All statistical tests will be two-tailed with a significance level set at $\alpha = 0.05$. Data analysis will be carried out using the R statistical software [41].

**QMRA.** QMRA is a formal risk assessment process (based on a modified chemical risk approach) that includes the classical steps of hazards identification and characterization, exposure assessment, and risk characterization. This process involves formulating and applying models using fields and literature data to represent a scenario while addressing uncertainties. QMRA models will be designed and applied to the four selected target viruses and for the different settings in order to assess the risk of ARI based on environmental contamination measured by the microbiological air and surface monitoring. The contamination values, along with the estimated frequency and duration of exposure through inhalation and contact obtained from questionnaires, will be used to calculate the dose, which is the amount of pathogen assumed in a unit of time. Probabilities of infection and disease will then be estimated by applying virus-specific dose-response relationships. These probabilities will be expressed as functions that take into account variability and uncertainties, using Monte Carlo Analysis applied through Vensim software. A sensitivity analysis will also be performed to identify the most influential variables [42].

## Outcomes of the study

The impact of ARIs is greatest in fragile and disadvantaged populations [43]. Our project has a clear public health focus: to reduce incidence of ARIs, limit their economic burden on the NHS, and address inequalities in disease burden related to age and marginalisation. While the protocol is developed and piloted within selected facilities in Tuscany Region, the methodology applied, and the findings can be generalised at national/international levels in similar settings. The expected outcomes corresponding to each specific objective are detailed below.

1. Sentinel clinical surveillance system for acute respiratory diseases will be established, with data collection tools (questionnaires) for clinical surveillance and risk factors tested and validated in different indoor settings such as LTCFs, prisons, and schools.

2. Indoor air quality assessment instruments and tools will be validated in the above-mentioned settings. Additionally, methods and strategies for assessing air and surfaces microbial contamination will be validated in the same settings. Methods and strategies for WBE will also be validated for the same selected communities.

3. An integrated environmental and clinical surveillance system will be established and tested for closed settings. This will involve:

   a. Assessing the connection between respiratory disease incidence, environmental and individual risk factors, and wastewater viral load. A predictive model using a nomogram will be developed, validated, and applied;

b. Validating a QMRA model for different viruses and settings;

c. Providing recommendations for expanding the integrated surveillance system of ARIs for early warning and pandemic preparedness efforts.

## Discussion

The Stell-ARI study is designed to develop and pilot a surveillance and management approach for ARIs within closed and vulnerable communities. By integrating environmental, behavioural and clinical data, it may be possible to provide the authorities with specific information for early detection and mitigation of ARIs.

To face the risk of ARIs outbreak onset and spreading in closed communities, different interventions can be adopted according to the natural history of the disease:

- Before a pathogen enters the community, assessing environmental conditions (using environmental questionnaires) and contamination (through surface and air monitoring), along with understanding people's behaviors and health status (via basal questionnaires), can facilitate a preliminary risk assessment. Quantitative Microbial Risk Assesssment (QMRA) models may be employed for this purpose. The results of this assessement can guide the adoption of preventive measures, even in the absence of the specific pathogen. These measures may include cleaning, ventilation, masks usage, and vaccination, The outcomes of the STELL-ARI project can identify proxies, indicators and index pathogens for an efficient, reliable, and sustainable continuous risk assessment, allowing the establishment of thresholds to recommend appropriate interventions.

- When a pathogen is first introduced into a community, efficient outbreak control relies on timely detection, ideally before symptomatic cases emerge. An effective early warning system can be established using wastewater-based surveillance (WBS), Which has already been extensively used for SARS-CoV-2 in communities [44]. The STELL-ARI project can demonstrate the potential application of this surveillance method for other respiratory viruses as well. Once an early warning signal is detected, reinforcing preventive measures becomes crucial. After the outbreak, the WBS can indicate the cessation of viral circulation.

- Finally, clinical and virological surveillance within communities enables the monitoring of symptoms and clinical outcomes, facilitating an evaluation of disease epidemiology and the effectiveness of preventive measures.

Overall, the project aims to establish a feasible and cost-effective approach to addressing uncertainties surrounding future viral outbreaks. By enabling rapid response and preparedness [45], it might provide, even with limited resources and time [46], an effective methodology for evaluating an acute viral spread and a long-term surveillance [47]. Numerous studies demonstrate the efficacy of predictive modelling that integrates clinical data and wastewater monitoring for tracking respiratory viruses' spread within communities and anticipating outbreaks [48, 49]. However, there is limited data on the potential combination of these predictive models with quantitative microbial risk assessment (QMRA), particularly in closed and vulnerable communities. QMRA offers a structured approach to evaluate the risk posed by infectious agents in various settings [50], but its integration with near-real-time surveillance systems remains underexplored. By combining predictive modelling with QMRA, it may be possible to identify and mitigate risks in these environments, ultimately leading to more effective prevention and control measures. The study's emphasis on combining predictive modelling with risk assessment could enhance our ability to detect and respond to outbreaks in these

environments before they reach critical levels, thus minimising their impact. Since this study involves different settings such as LTCFs, prisons and primary schools, it enhances the study's relevance and transferability across a spectrum of closed settings characterized by very different living conditions and population socio-demographic composition.

This study may have limitations. Coordinating data collection across such different settings presents logistical and coordination challenges. A complete correlation between clinical and environmental data could not be representative in case of limited data collection due to insufficient number of study subjects providing informed consent (e.g. refusal to undergo to nasopharyngeal swabs) or higher than expected loss to follow-up. This study potentially demands sustained participant engagement and ongoing data collection, which may discourage prospective participants, especially those in vulnerable situation, as well as parents of school children. Additionally, privacy concerns related to individual data collection and management, especially in a school environment, may reduce participants' confidentiality and trust. Another limitation lies in the large variability of environmental conditions and individual behaviours across the different settings analysed, potentially affecting the generalizability of the findings to other context. Finally, while innovative, reliance on advanced analysis techniques and biomolecular PCR tests could pose challenges in terms of expertise required for implementing this approach across various setups.

However, there are also several strengths. Since Stell-ARI's approach integrates environmental, behavioural, and clinical data, it enhances the accuracy and effectiveness of ARIs surveillance by capturing a multifaceted view of the factors influencing viral spread and providing a more complete understanding of transmission dynamics. This approach is valuable for identifying high-risk areas and populations, enabling continuous observation and rapid responses to emerging threats. It also facilitates early warning systems, proper resource allocation, and minimization of disease spread [51]. Tailoring interventions based on specific risk factors identified through comprehensive data analysis may lead to more effective prevention and control measures. Additionally, including behavioural data into the risk assessment tool provides insights into the impact of human actions and social behaviours on ARI transmission, aiding in the design of effective public health measures to reduce disease spread. The implementation of this comprehensive approach in challenging settings, such as prisons, schools, and long-term care facilities—where data collection is particularly difficult and scarce literature exists—adds value to the Stell-ARI project. Furthermore, the reproducibility of this integrated system into other settings is also a significant strength, since it can be adapted to various environments, including long-term care facilities, dormitories, schools, and prisons. In addition, Stell-ARI ensures participants privacy, respecting informed consent and pseudonymized data collection. This commitment will attest the study's credibility and minimise ethical concerns, encouraging participants in continued engagement in surveillance efforts.

The approach of the Stell-ARI project may offer several policy recommendations to enhance public health strategies and reduce ARIs. First, integrating data from various sources, such as clinical, behavioural, and environmental data, can facilitate the timely provision of real-time information to decision-makers. Second, implementing early warning systems in closed and vulnerable communities can help limit potential outbreaks by enabling rapid response and containment measures. Third, considering environmental and behavioural factors in the transmission and prevention of ARIs can encourage policymakers to prioritize public health strategies that improve indoor air quality. This might be through better ventilation, regular monitoring of enclosed spaces, and measures to reduce crowding in confined spaces.

Finally, the findings from Stell-ARI study may provide a blueprint for a more proactive and effective approach for preventing and managing ARIs in a timely manner, including for unknown threats. This comprehensive approach not only adds scientific value and practical

applicability in the field of ARI prevention by addressing critical gaps and deficiencies, but also serves as a potential model for public health policy interventions.

## Study status

The study began on December 1st, 2022, and is currently in data collection phase (second phase). During the first phase, all three facilities have been successfully enrolled, and the study design has been outlined. The study has been approved by the Bioethics Committee of the University of Pisa, on October 11, 2023, under protocol number 0136245/2023 Review No. 43/2023. In addition, the study has been registered on the European Union electronic Register of Post-Authorisation Studies (EU PAS Register) with the protocol registration number EUPAS106190. Individual enrolment of participants begun in November 2023.

**Ethical considerations and privacy.** The study has been approved by the Bioethics Committee of the University of Pisa, on October 11, 2023, under protocol number 0136245/2023 Review No. 43/2023. The investigators ensure that the study will be conducted in full compliance with the European legislation on clinical studies [Regulation (EU) No 536/2014] and its national transposition [DM 15 July 1997; D.Lvo 211/2003; D.L.vo 200/2007 and D.L.vo. 52/2019] and the principles of the Declaration of Helsinki in order to ensure maximum protection of the subjects involved. The principal investigator agrees that the study will be conducted in accordance with this protocol and Good Clinical Practice (GCP). The principal investigator has provided to the relevant Ethics Committee and Competent Authorities the study protocol and any related documents provided to the patient (Information Notice and Informed Consent Form). If changes to the protocol become necessary during the course of the study, the sponsor will submit an appropriate protocol amendment request to the relevant Ethics Committee, the approval process will align with the Committee's established procedures.

The project with the planned interventions and the administration of questionnaires at various times during the course of the study, will not have an impact on the health status of the participants. Respect for persons and human dignity, fair distribution of the benefits and burdens of research will be ensured, while protecting the values, rights and interests of study participants. The project will also consider the social, cultural and historical experiences of the patients involved. The research process will not perpetrate or endorse, directly or indirectly, any aspect of discrimination, stigma or exclusion. During the activities, the project will uphold the Avoiding Harm Principle: risks will be assessed with respect to their physical, social and psychological implications for participants.

In addition, participants are invited to take part in the project following the procedure outlined in the 'Method of administration of questionnaires' section. Obtaining informed consent from the participants involves informational sessions where explanation of the study's objectives and methods were carried out. Leaflets has been distributed in multiple languages to accommodate the diverse backgrounds of our participants. Actual participation starts only after individuals have signed the informed consent form, indicating their willingness to join. It is the responsibility of the investigators, or their representatives, to obtain informed consent from surveyed subjects (or their parents, for students) after they have been adequately informed about the aims, methods, expected benefits and foreseeable risks of the study. The investigators or designees also inform the participants that non-participation or discontinuation will not result in any harm or negative consequences.

Participation is entirely voluntary, and the signed informed consent forms will be kept by the respective facilities (LTCFs, Prison and Primary school) for the duration of the study.

**Communication and dissemination.** The project will be illustrated in the enrolled facilities, intermediate and final results will be published in peer reviewed journals, scientific

reports, presentations at conferences, seminars and symposia and will be used to inform public agencies and authorities about evidence-based strategies to link wastewater, air and surface matrices with respiratory health in closed communities, the data will be presented in an aggregated and anonymous manner and no information that could identify participants will be reported in any way. Intermediate and final results will be shared through scientific publications and academic conferences. Information material concerning the use of the results will be produced for relevant stakeholders, including the project's sponsor (Ministry of University and Research with reference to mission 4—Education and Research—of the National Recovery and Resilience Plan, NRRP).

However, the views and opinions expressed are solely those of the authors and do not necessarily reflect those of the European Union or the European Commission. Neither the European Union nor the European Commission can be held responsible for them.

**Role of the promoter and the investigators.** The role of the promoter and the investigator in the study is established in the technical annex of the Stell-ARI project, on the basis of which the financial resources corresponding to the planned commitment have been allocated. In detail, the promoter is responsible for developing the study design, the data collection tools and the analysis plan. The investigators are involved in the definition of the study design and are responsible for enrolment. Data collection will be performed by the investigators, commissioned facilities and, limited to schools, by the ISI Foundation via the Influweb platform. The promoter, with input from all investigators, is responsible for writing the scientific and dissemination reports. All investigators will be included in the authorship of the products resulting from this study.

**Ownership of data processing.** The University of Pisa, with headquarter at Lungarno Pacinotti 43 in Pisa, Italy represented by the acting Rector, and, exclusively for the school setting, in collaboration with the ISI Foundation, headquartered at Via Chisola 5 in Turin, Italy represented by the President.

## Acknowledgments

We thank all the staff of the enrolled settings for their cooperation in the implementation of the project: LTCFs (Alberto Meneghini, Maria Teresa Carracino, Giulia Del Punta, Elena Brugiati), prisons (Enrico Della Porta, Anna Santinami, Lisa Perugino, Santina Savoca) and primary schools (Rossana Condello, Germana Delle Canne, Claudia Bertani, Valeria Di Natale). We thank ISI foundation for their support providing Influweb platform and developing a dedicated part of the platform for our study: Giovanni Frigione and Matteo Delfino.

## Author Contributions

**Conceptualization:** Annalaura Carducci.

**Data curation:** Annalaura Carducci, Guglielmo Arzilli, Nebiyu Tariku Atomsa, Giulia Lauretani, Marco Verani, Francesco Pistelli, Lara Tavoschi, Ileana Federigi, Marco Fornili, Davide Petri, Tommaso Lomonaco, Claudia Meschi, Alessandra Pagani.

**Formal analysis:** Annalaura Carducci, Guglielmo Arzilli, Nebiyu Tariku Atomsa, Giulia Lauretani, Marco Verani, Francesco Pistelli, Lara Tavoschi, Ileana Federigi, Marco Fornili, Davide Petri, Tommaso Lomonaco, Claudia Meschi, Alessandra Pagani, Antonello Agostini, Laura Carrozzi, Laura Baglietto, Caterina Rizzo.

**Funding acquisition:** Annalaura Carducci, Marco Verani.

**Investigation:** Annalaura Carducci, Giulia Lauretani, Marco Verani, Alessandra Pagani.

**Methodology:** Annalaura Carducci, Guglielmo Arzilli, Nebiyu Tariku Atomsa, Giulia Lauretani, Marco Verani, Francesco Pistelli, Lara Tavoschi, Ileana Federigi, Marco Fornili, Davide Petri, Tommaso Lomonaco, Claudia Meschi, Alessandra Pagani, Antonello Agostini.

**Project administration:** Annalaura Carducci, Guglielmo Arzilli, Nebiyu Tariku Atomsa, Giulia Lauretani, Ileana Federigi, Alessandra Pagani, Caterina Rizzo.

**Resources:** Annalaura Carducci, Caterina Rizzo.

**Software:** Annalaura Carducci, Daniela Paolotti, Ciro Cattuto, Lorenzo Dall'Amico.

**Supervision:** Annalaura Carducci, Guglielmo Arzilli, Nebiyu Tariku Atomsa, Marco Verani, Ileana Federigi, Laura Carrozzi, Laura Baglietto, Caterina Rizzo.

**Validation:** Annalaura Carducci, Marco Verani, Caterina Rizzo.

**Visualization:** Annalaura Carducci, Guglielmo Arzilli, Nebiyu Tariku Atomsa, Giulia Lauretani, Lara Tavoschi, Marco Fornili, Tommaso Lomonaco, Caterina Rizzo.

**Writing – original draft:** Annalaura Carducci, Guglielmo Arzilli, Nebiyu Tariku Atomsa, Giulia Lauretani, Marco Verani, Ileana Federigi, Davide Petri, Alessandra Pagani.

**Writing – review & editing:** Annalaura Carducci, Guglielmo Arzilli, Nebiyu Tariku Atomsa, Giulia Lauretani, Marco Verani, Francesco Pistelli, Lara Tavoschi, Ileana Federigi, Marco Fornili, Davide Petri, Tommaso Lomonaco, Claudia Meschi, Alessandra Pagani, Antonello Agostini, Caterina Rizzo.

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
