## [Decision Letter · Decision Letter 0]

8 Jul 2024

PONE-D-24-08261Integrated environmental and clinical surveillance for the prevention of acute respiratory infections (ARIs) in indoor environments and vulnerable communities (Stell-ARI): ProtocolPLOS ONE

Dear Dr. Atomsa,

Thank you for submitting your manuscript to PLOS ONE. After careful consideration, we feel that it has merit but does not fully meet PLOS ONE’s publication criteria as it currently stands. Therefore, we invite you to submit a revised version of the manuscript that addresses the points raised during the review process.

1.‎ Discussion” section should be revised basically. It is not suitable for a manuscript. ‎The ‎manuscript needs more adequate discussion with supporting latest references. ‎

‎2.‎ In the discussion section, the potential limitation of the study should be highlighted and ‎in ‎the conclusion, novel insight should be clearly highlighted.‎ ‎

‎3.‎ It is also recommended to discuss and explain what should be the appropriate policies based ‎on the findings of this literature.‎

‎4.‎ Please make sure your conclusions and future perspectives section underscores the ‎scientific value-added of your paper and/or the applicability of your results. ‎

We look forward to receiving your revised manuscript.

Kind regards,

Sara Hemati

Academic Editor

PLOS ONE

Journal Requirements:

"The authors acknowledge support from the Lagrange Project of the Institute for Scientific Interchange Foundation (ISI Foundation) funded by Fondazione Cassa di Risparmio di Torino (Fondazione CRT). DP was supported by the VERDI project (101045989), funded by the Horizon Europe programme of the European Union. Views and opinions expressed are however those of the author(s) only and do not necessarily reflect those of the European Union or the Health and Digital Executive Agency. Neither the European Union nor the granting  authority can be held responsible for them. LD and CC were supported by Fondation Botnar."

"This project is funded by the National Government of Italy through the National Recovery and Resilience Plan (NRRP) 'Italy Tomorrow', which is part of the Next Generation EU (NGEU) programme. The NRRP aims to address the economic and social impacts of the SARS-CoV-2 pandemic crisis while addressing structural weaknesses in the Italian economy and promoting ecological and environmental transition.

The Tuscany Health Ecosystem (THE), coordinated by the University of Florence, aligns with the NRRP's focus on 'Life Sciences', fostering growth in Tuscany's life sciences sector through collaboration between public and private entities to address innovation needs. THE comprises 10 Spokes, with our study falling under Spoke 2 (Preventive and Predictive Medicine), led by the University of Pisa's Department of Biology, in collaboration with other departments.

In accordance with the NRRP, the Research and Innovation Programme "THE - Tuscany Health Ecosystem" has a 36-month eligibility period, commencing on 01/12/2022 and extending no later than 28/02/2026. Investment Line 1.5 allocates funding for the creation and/or strengthening of 12 Innovation Ecosystems (THE) across the national territory. This specific project falls under Spoke 2 (Preventive and Predictive Medicine), with project code I53C22000780001 with a budget of 1,308,657.86 €.

The funders (The Italian Government) had no involvement in the study design, data collection and analysis, decision to publish, or manuscript preparation."

Reviewers' comments:

Reviewer's Responses to Questions

**Comments to the Author**

1. Does the manuscript provide a valid rationale for the proposed study, with clearly identified and justified research questions?

Reviewer #1: Yes

Reviewer #2: Yes

2. Is the protocol technically sound and planned in a manner that will lead to a meaningful outcome and allow testing the stated hypotheses?

Reviewer #1: Yes

Reviewer #2: Yes

3. Is the methodology feasible and described in sufficient detail to allow the work to be replicable?

Reviewer #1: Yes

Reviewer #2: Yes

4. Have the authors described where all data underlying the findings will be made available when the study is complete?

Reviewer #1: Yes

Reviewer #2: Yes

5. Is the manuscript presented in an intelligible fashion and written in standard English?

Reviewer #1: Yes

Reviewer #2: Yes

6. Review Comments to the Author

You may also provide optional suggestions and comments to authors that they might find helpful in planning their study.

Reviewer #1: ‎1.‎ Discussion” section should be revised basically. It is not suitable for a manuscript. ‎The ‎manuscript needs more adequate discussion with supporting latest references. ‎

‎2.‎ In the discussion section, the potential limitation of the study should be highlighted and ‎in ‎the conclusion, novel insight should be clearly highlighted.‎ ‎

‎3.‎ It is also recommended to discuss and explain what should be the appropriate policies based ‎on the findings of this literature.‎

‎4.‎ Please make sure your conclusions and future perspectives section underscores the ‎scientific value-added of your paper and/or the applicability of your results. ‎

Reviewer #2: I have thoroughly enjoyed reading your manuscript and believe it holds significant potential for publication. Your detailed methodology, clear objectives, and comprehensive approach to integrating various data streams are truly commendable. This study has the potential to greatly enhance our understanding and management of ARIs in enclosed environments, making a valuable contribution to public health.

Congratulations on your excellent work, and thank you for your valuable contribution to the field. I have no additional comments or suggestions for improvement. I highly recommend this manuscript for publication.

7. PLOS authors have the option to publish the peer review history of their article (what does this mean?). If published, this will include your full peer review and any attached files.

Reviewer #1: No

Reviewer #2: No

---

## [Author Response · Author response to Decision Letter 0]

27 Jul 2024

Thank you for your insightful and constructive comments. We have addressed all the feedback from the reviewers and the editor in detail in the attached "Response to Reviewers" document.

---

## [Editor Report · Decision Letter 1]

6 Aug 2024

Integrated environmental and clinical surveillance for the prevention of acute respiratory infections (ARIs) in indoor environments and vulnerable communities (Stell-ARI): Protocol

PONE-D-24-08261R1

Dear Dr. Carducci%,

We’re pleased to inform you that your manuscript has been judged scientifically suitable for publication and will be formally accepted for publication once it meets all outstanding technical requirements.

Kind regards,

Sara Hemati

Academic Editor

PLOS ONE
---

## [Editor Report · Acceptance letter]

8 Aug 2024

PONE-D-24-08261R1 

PLOS ONE

Dear Dr. Atomsa, 

I'm pleased to inform you that your manuscript has been deemed suitable for publication in PLOS ONE. Congratulations! Your manuscript is now being handed over to our production team.

Kind regards, 

on behalf of

Dr. Sara Hemati 

Academic Editor

PLOS ONE